# The Particle Breakage Effect on Abrasive Wear Process of Rubber/Steel Seal Pairs under High/Low Pressure

**DOI:** 10.3390/polym15081857

**Published:** 2023-04-12

**Authors:** Ziyi Zhou, Qin Zhou, Kun Qin, Shuaishuai Li, Kai Zhang, Tongxin Yuan, Weihao Sun

**Affiliations:** 1School of Engineering and Technology, China University of Geosciences (Beijing), 29 Xueyuan Road, Haidian, Beijing 100083, China; 15528539030@163.com (Z.Z.);; 2Key Laboratory on Deep Geo-Drilling Technology of the Ministry of Land and Resources, China University of Geosciences (Beijing), 29 Xueyuan Road, Haidian, Beijing 100083, China; 3Consulting Center of China National Coal Association, Beijing 100120, China; 4Beijing Gang Yan Diamond Products Company, Beijing 102200, China

**Keywords:** abrasive wear mechanism, pressure, FKM seal, particles breakage, fracture characteristics

## Abstract

Pressure has a significant effect on rubber seal performance in the abrasive environments of drilling. The micro-clastic rocks intruding into the seal interface are prone to fracture, which will change the wear process and mechanism, but this process is not yet known at present. To explore this issue, abrasive wear tests were carried out to compare the failure characteristics of the particles and the variation wear process under high/low pressures. The results show that non-round particles are prone to fracture under different pressures, resulting in different damage patterns and wear loss on the rubber surface. A single particle force model was established at the soft rubber–hard metal interface. Three typical breakage types of particles were analyzed, including ground, partially fractured, and crushed. At high load, more particles were crushed, while at low load, shear failure was more likely to occur at the edges of particles. These different particle fracture characteristics not only change the particle size, but also the state of motion and thus the subsequent friction and wear processes. Therefore, the tribological behavior and wear mechanism of abrasive wear are different at high pressure and low pressure. Higher pressure reduces the invasion of the abrasive particles, but also intensifies the tearing and wear of the rubber. However, no significant differences in damage were found for steel counterpart throughout the wear process under high/low load tests. These results are critical to understanding the abrasive wear of rubber seals in drilling engineering.

## 1. Introduction

Abrasive wear of the elastomer seals is the most common type of failure that limits the lifespan of the equipment in the petroleum industry, mining, geotechnical engineering, farming, etc., and especially in drilling engineering [1,2]. Rubber seals are widely used as barriers to prevent rock fragments and debris generated during the drilling process from entering downhole tools and to protect drilling tool components from wear and damage. The elastic deformation of rubber transfers the pressure to the contact area on the shaft surface to prevent particles from entering. Therefore, the contact pressure between the seal and the shaft surface has a direct impact on the sealing performance of downhole tools or equipment [3,4]. Rock chips and debris can easily enter the sealing interface from the violent vibration of downhole tools, causing abrasive wear of the seals and shortening their service life [2,5]. Increasing the contact pressure of the rubber is a common method to reduce particle intrusion and abrasive wear in the drilling environment [6], but high pressure could aggravate the wear loss of the rubber seals. Typically, low contact pressure is widely used in shallow drilling to reduce friction and wear loss, but low pressure also makes it easier for fragments to enter into sealing interface [7]. In deep well drilling, high pressure is a common method to prevent debris intrusion. However, higher contact pressure can increase the friction and wear of elastomer seals and shorten the life of rubber seals [8,9,10].

Even with the shielding of the rubber seals, it is inevitable that some particles and debris will enter the sealing interface during the drilling process, resulting in severe abrasive wear [11]. Some studies have shown that the abrasive wear mechanism is directly related to contact pressure [3], but how the pressure affects abrasive wear behavior remains unclear. Dube and Hutchings classified three-body abrasion as “low-stress abrasive wear” and “high-stress abrasive wear” for different contact pressure [12]. High/low pressures lead to different forms of abrasive wear: three-body wear if the abrasive particles rotate between the surfaces; two-body wear if the particles are embedded on surface [13,14]. Three-body abrasive wear tests were carried out to compare the wear scars on the steel surface under the minimum and maximum applied load. It was found that with the higher load, the friction, and the wear loss aggravate, the wear mechanism changes from micro-cutting to micro-cutting and adhesive wear [15]. However, the above studies did not take into account the effect of particle breakage on the three-body wear process and wear mechanism.

In the drilling environment, the vibration of the drill string will cause debris to move into the sealing interface [7]. Abrasive particles also undergo squeezing and fragmentation process during the three-body wear, and these alter the wear process and damage mechanism. Several studies reported that the crushing of abrasive particles at the contact interface is affected by contact pressure. During relative motion, abrasive particles were crushed to varying degrees under high contact pressure [1]. The change in particle size affects the abrasive wear performance and the tribological mechanism of rubber sealing. [16,17]. Our previous study also found that the fracture of abrasive particles is very common in simulated downhole conditions, and the change process of abrasive wear is closely related to the continuous fracture process of abrasive particles [2]. The fracture of individual particles was analyzed and the damage mechanisms under different compressive loads were investigated [18]. The fracture characteristics and the variation process of particles under high/low pressure are still unclear. The effect of particle fracture on the tribological mechanism of rubber seals has also rarely been mentioned so far.

The three-body abrasive wear test was designed to investigate the features of particle breakage and its effects on the abrasive wear processes and mechanisms. In this experiment, the FKM ring was selected as a sample because of its chemical and heat resistance and for its common application in drilling engineering. A rotary seal pair of FKM/304 steel after a single entry with irregular SiO_2_ abrasive was tested to analyze the fracture mechanism of particles at the sealing interface and to classify the breakage characteristics of individual particle. By comparing the difference in continuous breakage processes of the abrasives under high and low loads, it was found that fracture types, degree, and breakage process of the particles had a significant effect on the removal mechanism of the rubber material. The present work provides insight into the disintegration characteristics of abrasive particles under different pressures and the abrasive wear processes on rubber/steel seal pairs. Our study provides useful guidance for the selection of sealing parameters in complex and harsh abrasive environments.

## 2. Experimental Procedure

### 2.1. Test Rig and Methods

Abrasive wear experiments were performed in a dry environment using the MMW-1 multi-function tribometer. The test rig was manufactured by Jinan OuTuo Test Equipment co. LTD (Jinan, China), which conformed to the standard of ASTM D3702-94. The stainless-steel (SS304) disc was fixed on the upper specimen holder and driven by an AC motor. The speed range is 0 rpm–2000 rpm. The FKM ring sample remained stationary on the holder (Figure 1).

Drilling holes were filled with micro-clastic rocks and detritus, which could easily enter the seal’s interface, causing abrasive wear. In order to simulate the abrasive wear behavior after particles invasion, the SiO_2_ particles were pre-spread uniformly on the FKM sample so that the particles were present in the interface at the beginning of the test. The test setup is shown in Figure 1. This method has been proven viable in previous studies [2,4]. The SS304 disc was driven by an electric motor with a sliding speed of 0.26 m/s [6]. Then, the FKM ring was pressed on the upper SS304 disc to achieve opposite sliding of the two surfaces. In this test, the compression rate of the FKM seal was converted to express the contact pressure of rotating rubber sealing. The compression rate of the FKM seal reaches 8% when the normal load reaches 100 N, which is commonly used in shallow drilling. This compression rate can be defined as a low-pressure seal. When the normal load increases to 250 N, the compression rate reaches 10%, which is more common in deep drilling and is also referred to as a high-pressure seal. High-pressure seal can prevent more debris from invading [19]. All tests were carried out at room temperature. The friction coefficient was recorded in real-time, and the mass losses of the specimen were measured by an electronic scale.

The test duration is 90 min, and a series of segmented tests were performed at the interval of 15 min to obtain the different interface states at different times during the abrasive wear process. To reduce the potential error, results were obtained by averaging three tests in each condition. The average values of mass loss were measured and recorded after three repeated measurements under the same conditions. The mass values were accurate to 0.01 mg by means of an electronic balance. After each wear test, the surface morphologies of worn tribo-pairs and the status of the particles were observed by SEM (Zeiss, Beijing, China).

### 2.2. Test Materials

The structures/settings of the rubber ring (FKM) and stainless-steel plates (SS304) in the test are shown in Figure 1a. The FKM ring has an outer diameter of 47.1 mm, an inner diameter of 40mm, and a thickness of 7 mm. The FKM, which was produced by Sannai New Materials Co., LTD. (Changshu, China), contains vinylidene fluoride (80 wt%) copolymerized with hexafluoropropylene (20 wt%). The detailed mechanical properties can be found in Table 1. The SS304 disc (Tianjin Lurui Stainless Steel Trading Company (Tianjin, China)) has an external diameter of 55 mm and a thickness of 7 mm. The composition of SS304 is shown in Table 2. The main physical properties of the SS304 include Poisson’s ratio (0.33), surface hardness (33 HRC), and density (7.93 g/cm^3^).

Prior to testing, the surfaces of the SS304 specimens were polished until the surface roughness (Ra) reached 0.05–0.1 μm. FKM specimens were cleaned with ethanol in an ultrasonic instrument and dried in the oven for 30 min at room temperature to remove moisture. The surface micrographs of specimens are shown in Figure 1e,f. Silicon dioxide (SiO_2_) particles were selected as abrasive particles because SiO_2_ is the main component of rocks present in many drilling applications. Most of the abrasive particles (nonuniform) have diameters between 5 μm and 500 μm in drilling [2]. In the experiments, the SiO_2_ particles with irregular shapes were selected as the abrasive particle to imitate the rock fragments, and the average particle sizes were approximately 100 ± 20 μm, as shown in Figure 1d. The particle had a Mohs hardness of 7 and a density of 2.2 g/cm^3^.

## 3. Fracture Mechanism of a Single Particle at the Sealing Interface

### 3.1. Fracture Types and Mechanism Analysis of SiO_2_ Particles

#### 3.1.1. Fracture Types and Mechanism Analysis of SiO_2_ Particles

After each tribological test, the broken abrasive particles that remained on the rubber surface were collected. Most of the fragments had irregular wedges and bulks, as shown in Figure 2, and the time-variable characteristics of the broken particle under high/low pressures were different. These different sizes of particle fragments were classified into three different size forms: ground, partially fractured, and crushed. They are marked with different symbols in Figure 2. The size of the fines was mainly within 10 μm and marked with an ellipse symbol, defined as ground damage form. The debris was a combination of a larger piece and some fines and marked with a hexagon, defined as partially fractured. When the abrasive particle was broken into small pieces of sub-particles and marked with a rectangle, those were defined as crushed.

Under high pressure (250 N load), particles in all three states (ground, partially fractured, and crushed) co-existed in the early stage, resulting in non-uniform particle sizes (Figure 2a,b). When the wear continued to the middle stage (45 min), more debris or fines were scattered around the particles (Figure 2c). More ground fines and partially fractured debris appeared around the particles when the wear process was stable (Figure 2d). At last, some debris was retained and scattered on the surface of the FKM (Figure 2e).

At low pressure (100 N load), the ground and partially fractured forms were found in the initial stage (Figure 2a-1,b-1). After 45 min of abrasive wear process, completely crushed and partially fractured particles coexisted. Due to the cracks or crevices developed during the process, one particle would shatter into several pieces (Figure 2c-1). These shattered fragments, with new fracture surfaces and sharp edges, are easily ground in the process of particle movement [18]. As the wear process entered the stable stage, more and more particles were found to be completely crushed and ground. After continued disintegration, the size of fragments decreased, and the number of debris reduced due to the escaping of abrasive particles [2]. A few fine particles were left and adhered to the rubber surface, causing adhesive damage to the rubber surface. (Figure 2e,e-1). The result revealed that the fragmentation modes (as shown in Figure 3) of abrasive particles can be divided into three categories: Ground, Fractured and Crushed. The load is a key factor for the different damage forms and breakage processes of the particle under the abrasive wear test. In the early wear stage, the partially fractured and ground were more common at a load of 100 N load, while more particles were crushed at a load of 250 N. When entering the stable wear stage, the particle size decreased continuously, and crushed particles were more difficult to observe.

#### 3.1.2. Fracture Mechanism of a Single Particle at the Sealing Interface

Rock cuttings have irregular shapes during the drilling process. Compared with the round shaped particles, irregular rock fragments are more susceptible to disintegration, and they could cause more damage to the surface of the FKM sealing [20]. When particles with irregular shapes enter the sealing interface as the third body, they are not only squeezed by rubber due to the viscoelastic deformation of rubber but also slide against the metal surface and lead to contact and extrusion of each other. Under these actions, particles would transform into different forms.

To better understand the experimental results, a single particle with polygon edges was selected as a typical sample to study the fracture mechanism, as shown in Figure 4a. Under the compression of the FKM ring, the particle between the hard steel and soft rubber was subjected to the combined forces of Ns, Fs, Nr, Fr, P, and Fp. Ns represents the normal force exerted by the steel disk, and Fs represents friction force from the steel disk, which is opposite to the movement direction of the particle. Nr is the concentrated force of the deformation stress of rubber acting on the contact area of the particle, and Fr is the friction force from rubber. P is the force exerted by adjacent particles on the particle surface, and Fp is the friction force from the adjacent particles. Therefore, the particle breakage was mainly caused by the forces of Ns, Fs, Nr, P, and Fp, while the movement of the particles was caused by Fs, Fr, and Fp.

The detailed fracture mechanism of the individual particles is depicted in detail in Figure 4b. In the case where the rubber is deformed by compression, the sharp edges of the particles easily penetrate and stick-shear the rubber surface. Meanwhile, the particle slid and rubbed against the metal surface under Fs force. Under small pressure, the deformation contact areas between the particle and rubber are relatively small, which indicates that the particle is sheared and ground under small Ns and Fs forces and is rarely crushed. Under larger pressure, the contact areas between particle and rubber enlarge, and the Nr and Fs forces increase. Further compression and shear may also lead to fractal flake or fine due to greater friction and shear forces, forming partial fracture [21]. The particles squeeze against each other, resulting in a tightly interlocked and embedded particle structure, which also intensifies the extrusion and surface sliding friction against the metal surface. All of these increases the forces P and Fp, which tend to lead to partial fracture of particles. When the compressive forces P and shear forces Fp are large enough, particles have partially broken into pieces of bulks or split [22].

Images in Figure 2 showed that three types of particle fractures co-existed under high load in the early stage, and more particles were crushed. With the increase of Ns, the contact deformation zone and the deformation force Nr may have also increased. For different shapes of particles, different Nr may have been aggravated the differences of the Fr, Ns, and Fs. At the same time, the force P of adjacent particles may have become more intensified with increasing Nr, and shear slip and dislocation may have become more intense, which lead to the particle being more susceptible to being broken. Therefore, higher forces could result in the coexistence of different fracture forms of particles and may also accelerate the crushing rate.

As friction and wear continue, the size of the particles decreases and the viscoelastic deformation area of the rubber and Nr, Fr, Ns, and Fs may decrease. For the remaining particles in the interface, some may have been damaged under the forces of Ns, Fr, and P, while others may have rotated and escaped from the interface under the action of Fs and Fr. On the other hand, the viscoelastic deformation and hysteresis deformation of FKM rubber may change with different sizes of particles, which may lead to different stress distributions on the particles. For some remaining particles under the action of cyclic deformation of the rubber, internal cracks expand and propagate, which may lead to crushing or splitting. Finally, the particles continue to break up and some fragments or debris may escape from the interface. The number of remaining particles decreases, and the particle size becomes too small for further fragmentation and may be left on the interface as three-body wear abrasive.

Despite the complex and random movement of particles at the sealing contact interface, three different breakage types could still be identified: ground, partially fractured, and crushed. It has been established that fracture form was affected by the size and shape of the particles. Most particles fractures occur near the edges and boundaries where shear breakage occurs. One hypothesis is that in more angular particles, less energy is required to break the particle because the sharper contact concentrates the stress on the particle [23]. The form of particle breakage between the soft-hard interface is related to the deformation degree of the rubber, the angle of the particles, and the size of the particles. The fractured mechanism and process will change the wear behavior of the rubber sealing.

### 3.2. COF Curve in Real-Time under High/Low Loads

The variation of the friction coefficient in respect to time reflected the process of particle breakage at 8% and 10% compression rates. The trends can be found in Figure 5.

It was noteworthy that the COF fluctuated drastically in the first 15 min under high pressure (250 N load), mainly due to the viscoelastic deformation of FKM, which increased the actual deformation stress and uneven contact pressures [15]. These lead to the co-existence of the different breakage types, including crushed, partially fractured, and ground. However, the COF at low pressure (100 N load) was smoother and lower than that at high pressure, which means that small deformation with rubber led to a smaller contact area, and a few crushed particles were found in the early stage. These results verified that high pressure leads to frequent and rapid particle disintegrations, turning those fragments into small debris [2,4].

### 3.3. Time-Variable Characteristics of Mass Loss of FKM/SS304 Pairs

The particle, as a 3-body, was compressed and fractured in different forms, which significantly impacts the mass loss of the FKM. There was minimal difference in the mass losses of the SS304 under low and high pressures. 

The mass loss of FKM is greater under high load conditions compared to low load conditions [1], as shown in Figure 6, and the total mass loss of the FKM sample was about 2 mg at 250 N (4~5 times higher than 100 N). At 100 N load, more particles were ground and partially fractured in stage I. Some SiO_2_ particles and metal debris were inserted into the rubber ring. The amount of mass loss of material was less than the combined mass of particles and debris. Therefore, a negative trend can be observed [19,24]. After reaching the peak at 45 min, the mass loss trend turned into stage II: the increasing stage. In this stage, the particles were constantly crushed and the surface material of FKM was removed due to the three-body abrasive wear. There are four stages under 250 N load. Rapidly increasing mass loss appeared in stage I and stage III. After stage III, there was little change in mass loss in Stage IV, which indicated that the wear processes had stabilized.

The wear amount of the SS304 seal pairs under different loads is shown in Figure 7. Some studies have shown that an increase in pressure leads to an increase in metal wear [25], and the two curves showed similar trends. The total amount of mass loss was almost the same at the end of the tests. However, there is a clear difference in the variation of the mass loss during the early 30 min of the wear process. The sharp increase of mass loss of SS304 in the early wear stage can be explained by the increase in the number of abrasive particles embedded between the sealing interface, especially on the softer rubber side, thus causing extra damage to the metal seal surface [26,27]. The mass loss of SS304 increased to 3.8 mg in 30 min at a load of 100 N, while the mass loss spiked to 3.8 mg within 15 min at a load of 250 N, and both wear amounts increased slowly and gradually as the wear progressed until a steady state of wear was reached. This implied that the breakage of the particles caused three-body abrasive wear of the sealing pairs, reducing the subsequent mass loss of the SS304 disk. The time-variable curves of the SS304 showed that the high pressure will accelerate the wear amount of SS304 in the early stage and reach the stable wear stage more quickly, but the pressure has little influence on the total wear amount.

These time-variable curves implied that the mass losses of the sealing pairs were closely related to the fracture and variation of SiO_2_ particles at the sealing interface. It is generally accepted that fragment size was directly related to mass loss of the steel, and larger particles have a greater effect on mass loss under high-pressure conditions [12]. Therefore, the mass loss of the SS304 increased faster under high pressure in the early stage. However, it is still unclear as to whether there is a correlation between the different fracture characteristics of the particles and the mass loss of FKM at different pressures. The following section analyzes the surface damage morphology of the FKM/SS304 pairs.

### 3.4. Wear Topography Variation of the FKM/SS304 Tri-Pairs

#### 3.4.1. Wear Topography Features of the FKM

The wear morphologies of the FKM were divided into three stages, which displayed different features at high/low load (Figure 8). At 250 N load, some tear scars and micro-cutting pits were uniformly distributed on the worn FKM surface (Figure 8a). The micro-cutting pits were mainly caused by some particle edges piercing and shearing the rubber surface. In stage II, micro-cutting pits became deeper and many irregular cavities also started to emerge (Figure 8b,c). Finally, many smaller pits were scattered on the smoother surface (Figure 8d,e). These pits were the traces left after the micro-cutting pits were worn off. At 100 N load, a series of micro-cutting pits and some indentations appeared on the rubber surface in stage I. (Figure 8a-1,b-1). As the test progressed, the worn surface gradually showed more ploughing, which is parallel to the sliding direction (Figure 8c-1,d-1). In the final stage, some small typical “Schallamach waves” patterns gradually appeared (Figure 8e-1), which was a typical two-body wear morphology.

The dominant wear patterns of the FKM were Indentation and ploughing under 100 N load, while tearing and micro-cutting pits were predominant under 250 N load. Under 100 N load, indentation was the plasticity deformation of the FKM ring under continuous compression, which was the permanent and irrecoverable deformation. It showed that the permanent deformation of rubber was accompanied by network destruction and the relative flow of the molecular chain, which is an important viscoelastic behavior of the rubber. When the particles were wrapped by rubber, the sharp edges slid against the metal surface and cut some debris. These metal debris easily penetrated or inserted into the rubber surface during friction. Combined with the embedding of the SiO_2_ debris, the rubber mass loss showed a negative growth trend (Figure 6), as described in our previous research [2]. As the friction processes continued, serious ploughing appeared, indicating that the fresh fracture flakes were easily inserted and ploughed on the FKM surface. The ploughing trajectory transited from discontinuous straight line to line-point, implying that more fragments became smaller and that the motion changed from sliding to rolling. In contrast, under 250 N load, the particles were compressed under larger deformation forces, resulting in stress concentration at sharp edges and pitch-tearing of the rubber surface. As some of the original particles were crushed and fractured, smaller fragments were formed, and more sharp edges or corners stuck and micro-cut the rubber surface. The micro-cuttings pits became deeper and denser.

#### 3.4.2. Wear Topography Variation of SS304

Figure 9 shows a detailed analysis of the typical patterns of SS304 at different stages. The steel presents more significant surface damage under a load of 250 N. These damages show signs of microcutting and microploughing [17]. At about 15 min a slight microploughing started to appear (Figure 9a). Then, the microcutting signs decreased, while microploughing appeared more significantly on the abraded surface (Figure 9b). The microploughing became deeper and longer, which was thought to be caused by trapped particles and sliding movements on the soft rubber surface [27] (Figure 9c). Finally, the surface became smoother by microploughing (Figure 9d).

Under a load of 100 N, some microploughs started to appear (Figure 9a-1), showing very smooth line grooves caused by particles embedded in the rubber surface [28], and many microcuts started to appear at 30 min (Figure 9b-1). Following that, microcutting marks on the surface were alleviated and some slight signs of rubber adhesive appeared on the metal surface (Figure 9c-1). Adhesive wear became the dominant surface pattern in the final stage (Figure 9d-1).

Both microcutting and microploughing are more obvious wear patterns of SS304 under high pressure. This is due to the coexistence of the three breakage types of particles in the previous stage, where more particles are crushed by higher deformation forces and new angled fracture surfaces are stuck and sliding in the soft rubber. At a load of 100 N, the wear surface patterns were mainly moderate microcutting with less adhesive wear, mainly due to the smaller deformation force of the rubber and the smaller size of the debris.

## 4. Fracture Mechanism of a Single Particle at the Sealing Interface

From the wear topography variation of the FKM/SS304 tri-pairs, the breakage processes and fracture types of the particle had a direct effect on the wear damage morphology and wear mechanism of the sealing pairs [2]. Irregularly shaped particles are prone to fracture, which not only reduced the particle size but also decreased the deformation force of rubber. This changed the forces acting on the particle between the soft rubber and hard steel and altered the motion of the particles. Thus, the complex dynamic of friction and wear behavior was directly related to the processes of particles being broken, migrated, and escaped. The wear processes and transformations were compared at low/high loads (Figure 10a).

For the high pressure (250 N load), the particles were severely compressed by higher elastic deformation force. The high deformation force increased the extrusion and friction between particles and aggravated the uneven distribution of the stress on particles, resulting in the co-existence of different fracture types in the first stage (Figure 10b). Some larger fragments with sharp edges penetrated and tore the rubber surface, sliding on the steel surface to form microcracks and slight micro-ploughs. Following that, the deformation pressure of the smaller size particles decreased. These particles had many new fracture surfaces, which were formed by intense damage in the initial stage. The sharp edges of these fractures easily penetrated the soft rubber and slid on the surface of SS304, leading to the typical micro-ploughing. As the wear continued, the size of the chips became smaller and the force Nr decreased. Meanwhile, under the action of Fs and Fr, the motion of some pieces changed from sliding to rolling and some debris rolled away from the interface. Finally, many fines remained in the sealing interface as the third body, forming three-body wear. The wear mechanism changed from abrasive wear and sliding wear to three-body wear.

For the low pressure (100 N load), the abrasive particles were forced on the rubber to slide on the steel surface, resulting in micro-ploughing on the steel surface in the initial stages. Under rubber compression stress and friction of the metal, some cracks started to appear in the contact area of the particle and extend to the surface (Figure 10c). The wear mechanism in the first stage was mainly sliding wear [24]. The continuous deformation of FKM promotes the generation of cracks and fissures of particles during the friction process. These fragments with new fresh cutting or sharp edges pierced into the soft rubber and slide-rolled on the interface under the friction force, causing more ploughing marks on the surface of the FKM. At last, as friction force Fr decreased, debris escaped from the interface more easily and faster, leaving significantly fewer particles on the interface than under high pressure. The “Schallamach” wave and adhesion appeared, which was a typical wear scar of two-body stick-slip. Therefore, the transformation of the wear mechanism can be interpreted as sliding wear, sliding and rolling wear, and two-body wear.

The differences in abrasive wear mechanism are related to differences in particle breakage types, fragment size, and motion. The fragmentation and motion of irregularly shape particles are more complex than those of round particles. In the early stage, particle breakage and pressure have a significant effect on the wear mechanism. Particles with edges are more concentrated under high load, which led to more imbalanced rubber deformation and the coexistence of three fracture types. These new fracture edges and fresh sections pierced and tore the FKM surface, resulting in more severe wear of the rubber under high load. As the wear process continued, the size and movement of fragments influenced the subsequent wear behavior, resulting in significant differences in wear mechanisms under low and high pressures. Under low pressure, the embedding depth of abrasive particles decreased and the motion of abrasives changes from sliding to rolling, while under high pressure, the motion of abrasives tended to slide under large elastic deformation pressure. In the final stage, the escape of the particle and the number of debris remaining at the interface are the main factors that determine the wear morphology. At low load, due to the low deformation force of the rubber, the particles roll and escape faster, leaving few particles on the interface and increasing the possibility of adhesive wear. However, at high load, more detritus remains on the interface, forming three-body abrasive wear, and rubber wear tended to stabilize while metal wear increased slowly.

The pressure could be the main possible explanation for the wear mechanism transformation in abrasive wear. It has also been suggested that these results will provide experimental support for further investigation of the abrasion or particle invasion of rubber seals in complex drilling environments [29]. However, the abrasive wear mechanism of rubber is a complex process that is impacted by many factors, including particle breakage rate, elastomer deformation of rubber, and the changes in actual contact pressure. These results do not provide a complete explanation for abrasive wear. Further investigation of particle size and fragments distribution is recommended in order to draw the most accurate explanations.

## 5. Conclusions

This study reveals the influences of pressure on the particle breakage and on friction and wear processes of FKM in an abrasive environment. A series of abrasive wear tests were performed on FKM/SS304 sealing pairs with SiO_2_. The relationships between different forms of fragmentation and transitions of wear processes under high and low pressures were discussed in detail. The main conclusions are drawn as follows:The micro-clastic rocks intruded into the seal interface are prone to fracture during abrasive wear. A force model of individual particle at the soft rubber–hard metal interface was developed. The fracture mechanism was analyzed, and three types of fragments were described, including ground, partially fractured, and crushed. It was considered that the particle breakage is the main product of the forces Ns, Fs, Nr, and P, while the movement of the particles is the result of Fs, Fr, and Fp. The higher the pressure, the higher the stress and the faster the breakage developed, and the more obvious the coexistence of the three types of fragments.These transitions of the tribology behavior and wear mechanism were closely correlated with the different particle fracture characteristics. With the continuous breakdown of abrasive particles, the fragment size decreased, and the movement of the abrasive and subsequent fracture processes also changed. At low pressure, the wear mechanism changes from sliding wear to sliding and rolling wear, and finally to two-body wear. At high pressure, the change process is abrasive wear, sliding wear, and three-body wear.Compared to round particles, irregularly shaped particles are more easily crushed and fractured, forming more new sharp edges. These cuttings and edges can pierce, tear, and micro-cut the FKM surface, exacerbating the tearing and wear of the rubber under high pressure. However, the whole mass loss of the steel is similar under low and high pressures. Therefore, improvement of the surface hardness for rubber and steel can reduce the damage caused by particle penetration under high pressure and enhance the service life of rubber in abrasive conditions. These results provide experimental support for the appropriate selection of sealing pressure and structure design of the rubber in drilling engineering.

## Figures and Tables

**Figure 1 polymers-15-01857-f001:**
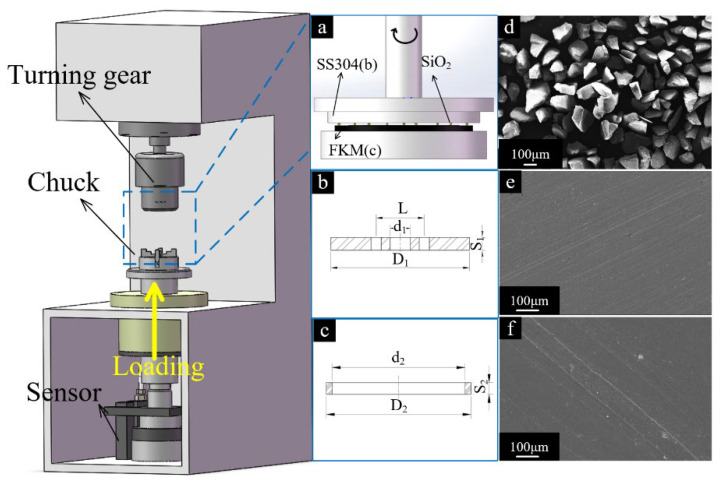
Schematic descriptions of (**a**) sliding friction pairs, (**b**) SS304 disc, (**c**) FKM ring, and SEM micrographs of (**d**) SiO_2_ particle, (**e**) metal disc, and (**f**) FKM ring.

**Figure 2 polymers-15-01857-f002:**
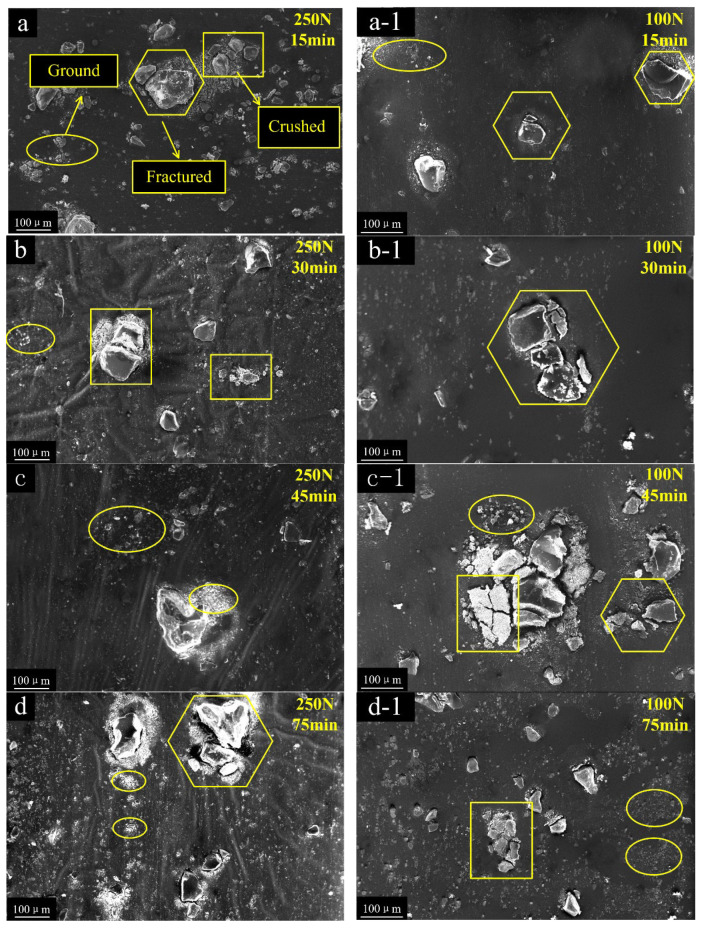
The changing process of abrasive particles from 15 min to 90min under the load of 250 N (**a**–**e**) and 100 N (**a-1**–**e-1**).

**Figure 3 polymers-15-01857-f003:**
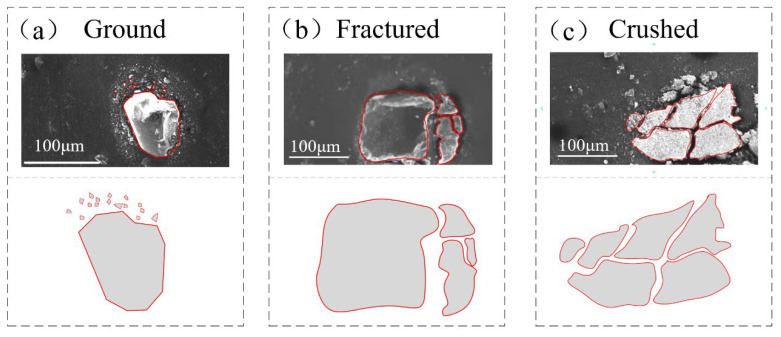
Typical particle breakage forms of (**a**) ground, (**b**) fractured, and (**c**) crushed.

**Figure 4 polymers-15-01857-f004:**
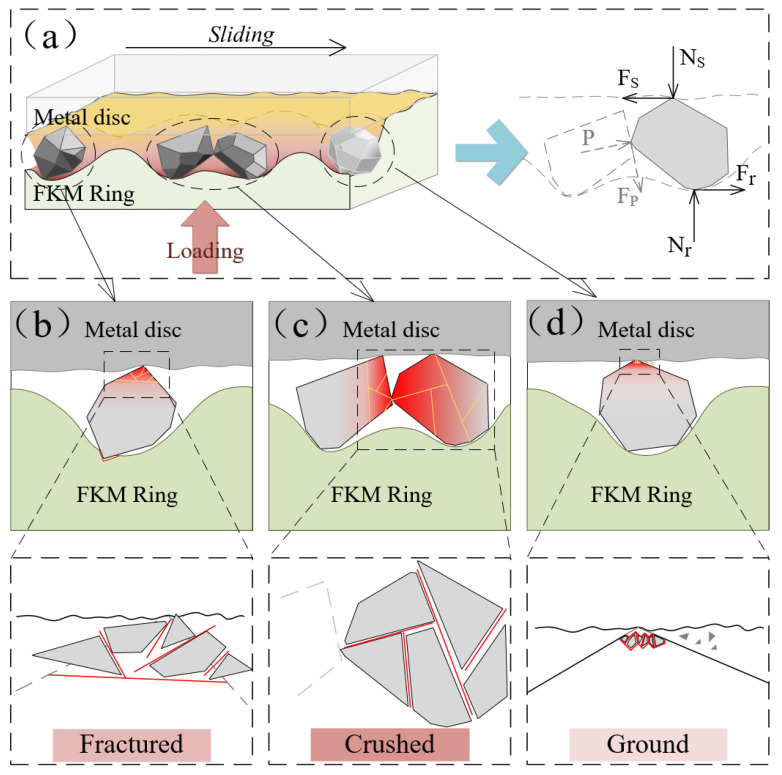
Schematic illustrates (**a**) the force analysis of abrasive particles in the three-body contact interface, and different particle breakage forms of (**b**) fractured, (**c**) crushed, and (**d**) ground.

**Figure 5 polymers-15-01857-f005:**
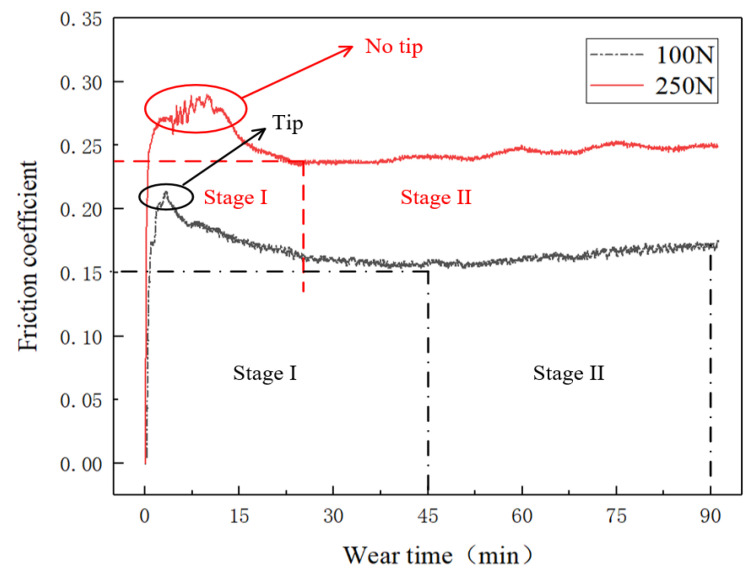
COF curves in the function of time under 100 N and 250 N load.

**Figure 6 polymers-15-01857-f006:**
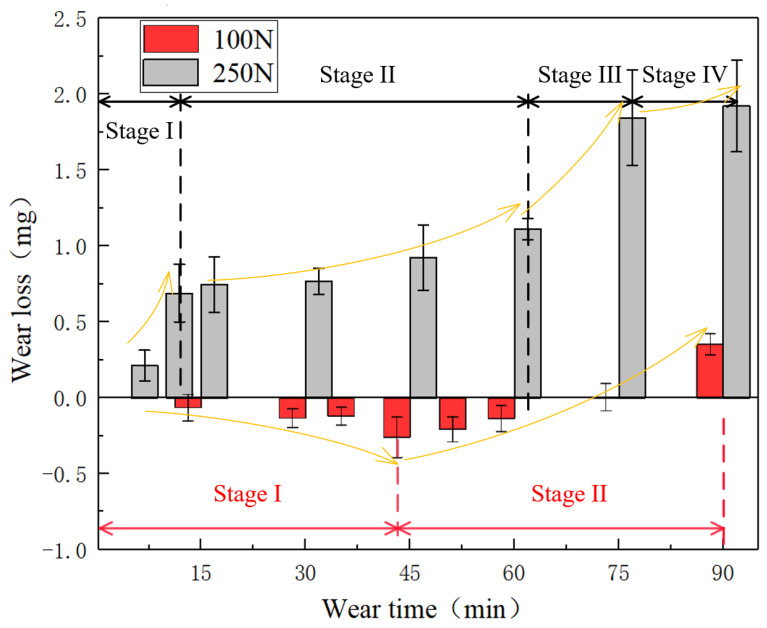
Time-variable mass loss of rubber seal under the load of 100 N and 250 N.

**Figure 7 polymers-15-01857-f007:**
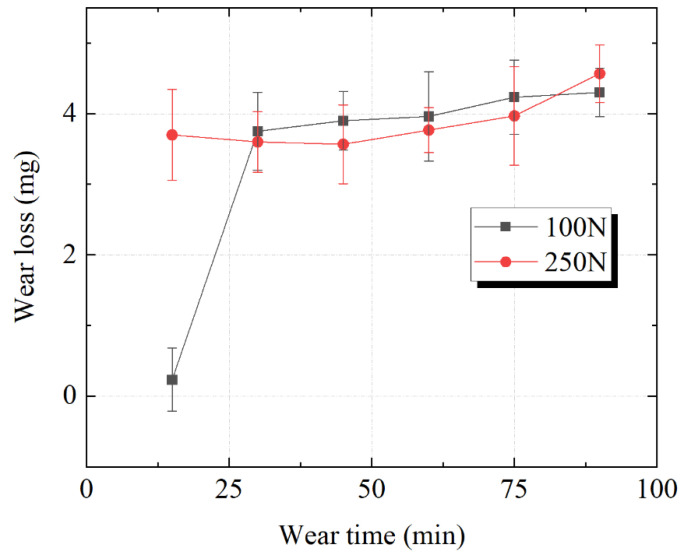
Time-variable mass loss of SS304 under the load of 100 N and 250 N.

**Figure 8 polymers-15-01857-f008:**
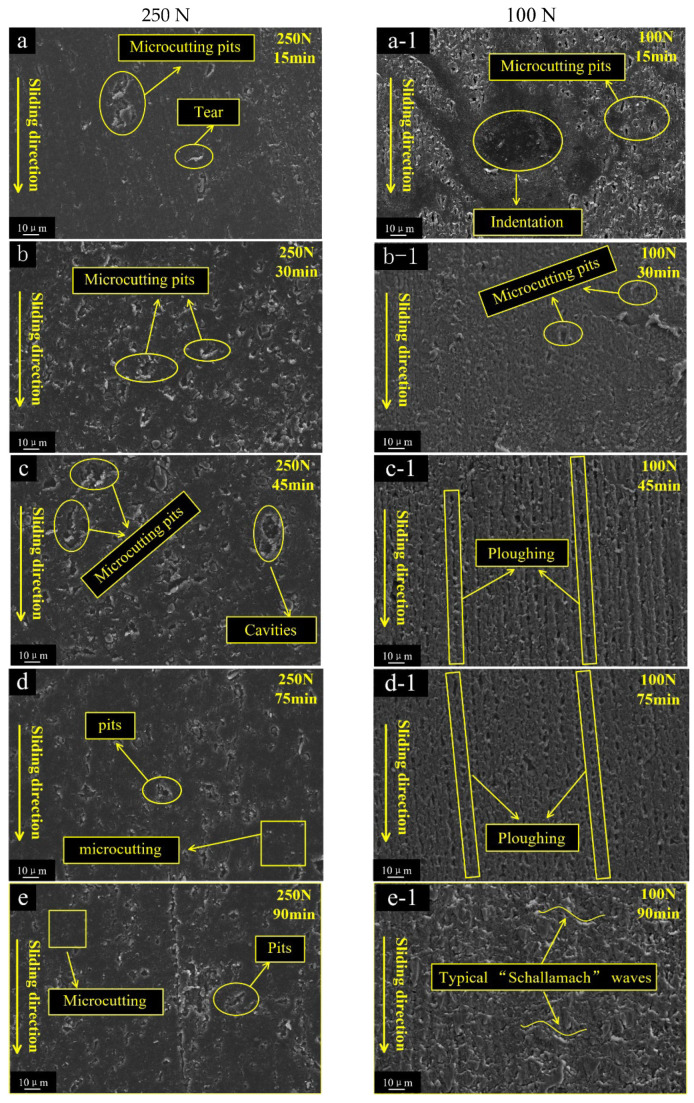
Wear topography variation of the rubber seal from 15 min to 90 min under the load of 250 N (**a**–**e**) and 100 N (**a-1**–**e-1**).

**Figure 9 polymers-15-01857-f009:**
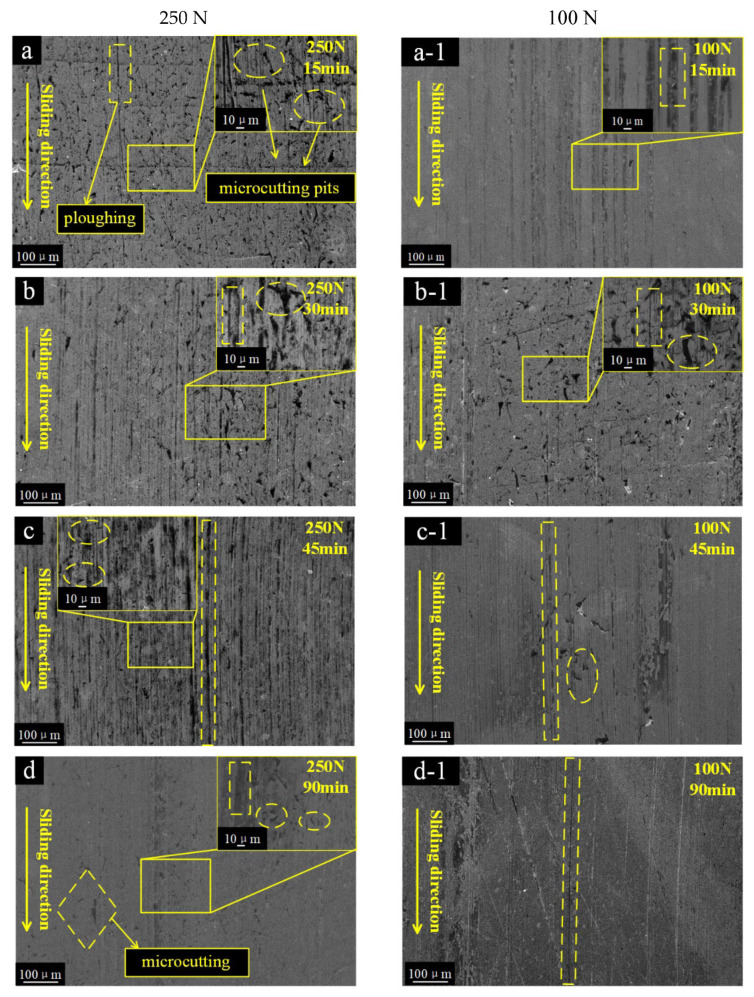
Wear topography variation of the metal disc from 15 min to 90 min under the load of 250 N (**a**–**d**) and 100 N (**a-1**–**d-1**).

**Figure 10 polymers-15-01857-f010:**
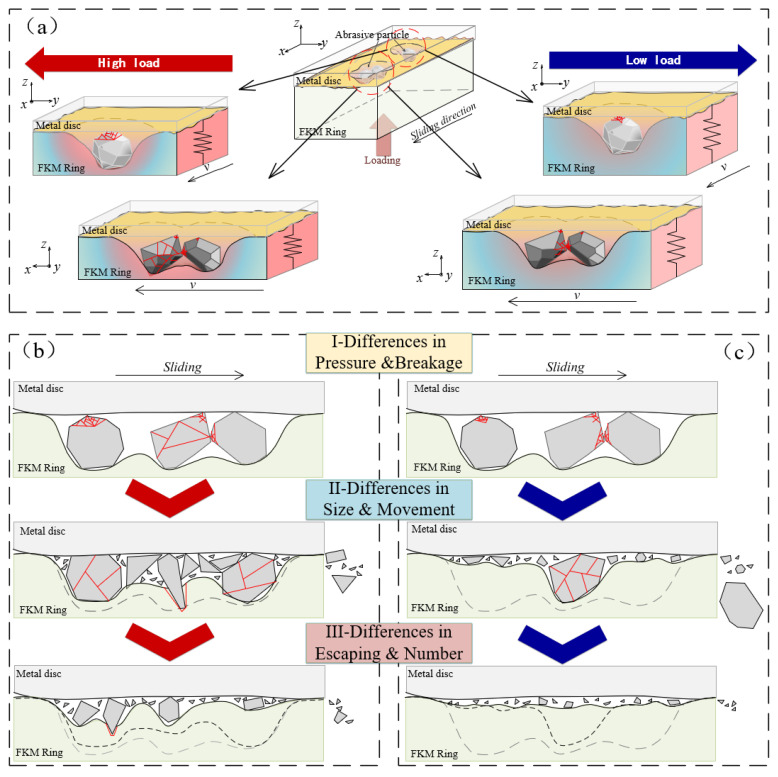
Schematic diagrams of (**a**) three-body contact of sealing interface under high/low pressure and the wear mechanism under (**b**) high pressure and (**c**) low pressure.

**Table 1 polymers-15-01857-t001:** Mechanical properties of FKM specimens.

Hardness	Density (g/cm^3^)	Tensile Strength (MPa)	Poisson’s Ratio	Elongation at Break	Elasticity Modulus (MPa)	Roughness (μm)
70 (Shore A)	1.85	16.8	0.48	300%	7.8	1

**Table 2 polymers-15-01857-t002:** The composition of SS304.

	C	Mn	P	S	Si	Cr	Ni	Fe
SS304	0.08%	2%	0.045%	0.03%	1%	20%	10%	66.845%

## Data Availability

Data sharing not applicable.

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
