# Peer review of "The Particle Breakage Effect on Abrasive Wear Process of Rubber/Steel Seal Pairs under High/Low Pressure"

_polymers, 2023, doi:10.3390/polym15081857_

Round 1

Reviewer 1 Report

Line 51. “The abrasion is similar to the wear damage caused by three-body wear”. The abrasion can be divided into abrasion of 2 and 3 bodies. They are not different phenomena. Line 57. “insert into the surface” Should be “particles are embedded on surface” Line 69. “The change in the particle size changes 69 the abrasive wear performance and the tribological mechanism of rubber sealing [16, 17].” The idea is not very clearly understood Line 100. It is not very clear how the particles were added to the contact. How much? Line 100. It would be interesting to know more about some mechanical properties of the particle. Line 103. Why was that speed chosen? Line 138. It would be interesting to know the profilometry of the surface of each of the specimens before and after the tests. Linea 153. Apart from the visual criteria, another technique was used to classify the sizes of particles and their fragments. Line 174. It would be worth having an electron micrograph to see how the adhesive wear was. Line 243. What other indications could you comment to ensure that there could be an abrasive wear of 3 bodies. Line 276. Is there evidence of debris inserted into the elastomer? (EDS) (SEM) Line 277. The term "mass loss" is more common instead of material Line 298 & 306. Repetitive ideas Line 311. Label Figure 7 is equal to label Figure 6. Image 8c-1 & 8 d1. I would consider that the marks could be tearing traces or rolling track Line 351. It would be interesting to observe the microcraking Line 352. The idea is not very clear and could be contradictory. Line 359. “Ssome”. It should be "some" Line 404. It might be better to express. The continuous movement promoted….

Reviewer 2 Report

1. Sentences start with "we......" should be avoided.

2. Reference no. 12,15, 20, 24 and 28 must be rewritten due to inconsistency with others.

3. Weight loss should be used instead of wear loss.

4. Wear rate should be calculated to verify stage of wear.

5. Material characteristics are not fully declared.

6. Quantity or concentration of SiO2 used in all tests are missing?

7. Roughness and hardness of FKM are not clearly and properly declared.

8. Sliding direction of Figure 9 must be added.

9. In three body abrasion, the harder surface (SS304 in this case) normally be abraded severely by hard contaminants? 

10. Is this work simulated properly for the real world problems? i.e. material of real drilling process, general rock hardness, applied load or pressure etc. Does the worn surfaces of SS304 in this work correlates well with the real world problem in particular?

11. Figure 1 must be improved to a clearer view specially SiO2 and original surfaces of both seal and SS304 specimen.

12. What was the sliding velocity in all tests.

Reviewer 3 Report

The subject of this work is important, and its study has great potential. There were good observations and images presented, for instance, Figures 2 and 3, and the representation of the fracture mechanism of a single particle at the sealing interface was interesting. Friction and wear graphics on Figures 6 and 7 presented some good insight on the particle breakage effect on wear of rubber/steel seal pairs interface. However, minor revisions need to be done in this manuscript, for it to be more concise and robust. These revisions include methodological issues, grammar and sentence structuring.

Please find attached a PDF document with all the methodological, theoretical and grammar/sentence structuring comments and observations made. Here are some of the main comments:

- The value of the SS304 steel hardness (HRC) is too high. It is necessary to check this value or specify why this value is out of the common range.

- No metric or criteria was mentioned whatsoever to justify the conclusion on which particle damage form was more present in each test condition/duration. The images presented in Fig. 2 (1 per test duration and condition) are about 600 micrometers wide, and 1 image per test condition/duration by itself cannot represent the whole debris' damage scenario. If there was any kind of statistical treatment or measurement made, this needs to be stated.

- The verb tenses in the simple past (i.e. ‘'increased', aggravated', 'became') in section 3.1.2 indicates that the conclusions were drawn out of observations and measurements of forces and displacements. However, it seems that this is a proposed mechanism/model for analyzing forces acting on a single particle. If it is indeed a proposed mechanism, and not experimental observations (in this case, forces and the displacements caused by them would have to have been measured), it is necessary to rewrite this passage, using verbs in the form of 'may have been aggravated', 'may have been increased', 'may have become', 'may have led to'. Another approach would be to write “"By assuming this proposed model, it can be said that …” and then use the verb tenses in the simple past.

- Some inconsistencies were noticed between what is shown in the images and the discussion of Fig 9, in section 3.4.2 of the manuscript.

- SEM images in Fig. 9 could be in a better resolution and bigger.
